

# CYCLIM: a semi-automated cycle counting tool for palaeoclimate reconstruction

Edward C. G. Forman[1], James U. L. Baldini[2]

[1]School of Geography and Environmental Science, University of Southampton, Southampton, SO17 1BJ, United Kingdom
[2]Department of Earth Sciences, University of Durham, Durham, DH1 3LE, United Kingdom

*Correspondence to*: Edward C. G. Forman (E.Forman@soton.ac.uk)

**Abstract.** Counting annual-scale fluctuations, such as geochemical cyclicity or visible growth bands, within a climate archive can yield extremely high-precision chronological models. However, this process is often time-consuming and subjective, and although various software packages can automate this process, many researchers still prefer to count manually given its

technical simplicity and transparency. Here we present a new tool that combines the time saved by automation with the flexibility afforded by expert judgement. CYCLIM uses a matched filtering approach to detect cyclicity and then allows the user to inspect and refine the automated output whilst also quantifying age uncertainty. The presented framework speeds up cycle counting by automating the first-pass of the count while also retaining the benefits of a manual count by allowing for post-analysis tuning. Across three examples using published palaeoclimate reconstructions, the automatic output found 96.2%

of the cycles, with a false positive and false negative rate of 3.3% and 3.8%, respectively. This means that only 7 cycles per 100 need to be corrected manually, making cycle counting with CYCLIM ~14.1 times faster.

## 1 Introduction

Palaeoclimate reconstructions are the principal method to put modern climate trends within a longer-term variability. These techniques use proxies recorded in natural archives to track past changes in the climate system, and so one of the key steps in

developing palaeoclimate reconstructions is translating depth (or growth) to the time domain. Chronologies are derived using a variety of techniques such as radiocarbon or uranium-series dating, but usually they can only supply a limited number of points because of sample material, analytical costs, and time constraints (Breitenbach et al., 2012). Cycle or layer counting, although relative, offers a powerful alternative, often reaching sub-annual precision (Comboul et al., 2014; Forman et al., 2025; Ridley et al., 2015).

Rhythmic layering and/or cyclicity in palaeoclimate archives can reflect seasonal to annual environmental variability, which can serve as the basis for a chronology. Cycle counting involves identifying and quantifying these repeating visual (e.g., colour or texture) or geochemical (e.g., stable isotopes or trace elements) parameters. However, prior to performing a count, the annual-scale fluctuations need confirmation as truly annual both with independent chronological validation and a theoretical basis for the processes involved in generating the periodicity. Many natural archives are amenable to cycle counting methods,



including: (1) tree rings by preserving annual growth rings (e.g., Anchukaitis and Evans, 2010); (2) corals through density
bands and geochemical signatures (e.g., Goodkin et al., 2005); (3) ice cores via multiple records such as dust, conductivity,
and stable isotopes (e.g., Winstrup et al., 2019); (4) lake/marine sediment through visible varves (e.g., Wick et al., 2003); and
(5) speleothems via physical laminae and geochemical cyclicity (e.g., Tan et al., 2006). Whereas cycle counting is prone to
error via missing/ambiguous cycles in addition to lacking the absolute dating precision afforded by some radiometric methods,

it offers unparalleled temporal resolution in many contexts and can augment existing chronologies, particularly when anchored
to a known age.

The process of cycle counting is often time consuming and subjective, so many studies have presented analytical tools to
automate the process. Some archives have purpose-built chronological packages due to the nuances of the systems. Tree ring
chronologies typically combine records from multiple trees using techniques such as cross-dating to overcome inaccuracies

arising from rings in one record that are missing or false (St. George, 2014). Similarly, ice core chronologies may require
correcting for thinning and compaction using techniques such as thinning functions and ice flow models (Kahle et al., 2021).
For this reason, archive-specific tools such as dplR (Bunn, 2008) and the hidden Markov models (HMMs) algorithm presented
by Winstrup et al. (2012) exist for dating tree rings and ice cores respectively, although, if the dataset permits, general tools
could still yield accurate age models for these records. These general tools are either applied to images/scans of the archive to

count visible layer boundaries or to depth profiles extracted from the archive to count detectable cyclicity. A degree of overlap
exists between these two methods because images/scans can convert into intensity profiles such as greyscale or fluorescence
to then undergo cycle detection using similar signal processing techniques to those of non-image-based tools. Visual packages
exist for counting both varved sediments (e.g., Weber et al., 2010) and speleothems (e.g., Sliwinski et al., 2023) using
techniques like machine learning. Signal processing algorithms simply use a depth-proxy record and identify cycles via

statistical techniques like peak detection, making them very versatile. Smith et al. (2009) developed a trace element cycle
counting algorithm that uses an estimate of mean cycle width obtained by spectral and wavelet analysis. The algorithm also
uses two threshold criteria: (1) a minimum cycle amplitude, fixed to a local standard deviation of signal variance; and (2) a
minimum inter-cycle separation, calculated relative to the preceding detections. Nagra et al. (2017) devised a multi-proxy
approach for counting annual trace element cycles in speleothems. Each trace element transect is normalised and concatenated

before undergoing a principal component analysis (PCA), after which principal component 1 (PC1) is passed to a peak counting
algorithm. This algorithm uses a prior estimate of annual growth rate as a sampling window and sets thresholds relative to the
mean cycle for determining whether a detection is false or missing. However, many researchers still opt to count by inspection,
partly because it offers greater flexibility and because the variability among proxy records can impede the effectiveness of
automated techniques.

Combining the advantages of both automation and expert judgement, here we present a Python-based application (CYCLIM)
that automatically detects cyclicity before allowing user-guided refinement of the results. Cycles are detected using matched
filtering with a Gaussian kernel and sections of ambiguous cyclicity are identified. Within the interface the user can then
review the output and adjust boundaries by removing spurious detections and/or adding in new cycles. Additionally, the user





can quantify algorithmic uncertainty using a noise perturbation-based Monte Carlo approach. This methodology provides a flexible and reproducible framework that streamlines the process by automating the initial detection phase while preserving the control necessary for dealing with the heterogeneity and noise of proxy records. By facilitating both efficiency and supervision, CYCLIM enhances the speed, accuracy, and transparency of cycle counting.

## 2 Interface and Methods

### 2.1 Graphical User Interface (GUI) Overview

The CYCLIM algorithm (Fig. 1) uses a semi-automated approach whereby the user can optimise the algorithm by tuning its hyperparameters and enabling additional features, before allowing inspection and refinement of its output. First, the interface prompts the user to import the depth-proxy dataset for counting. The depth-proxy dataset should: (1) have sufficient signal-to-noise to distinguish annual cyclicity from background noise; (2) exhibit no substantial or consistent change in cycle length; (3) have approximately equally spaced datapoints; and (4) ideally contain no missing values. The algorithm can tolerate a few missing values provided they are removed prior to counting and maintain a near-perfect consistency in sampling rate. The user specifies an estimate of the average cycle length and then can choose to use the automatic hyperparameter values or set them manually. Because the algorithm uses the average cycle length to calculate the template width, which has built-in flexibility, this only needs to be an estimate. Additionally, the algorithm can tune detected cycles to local minima and divide gaps using the expected cycle length. Post-detection the user can inspect the cycles boundaries that the algorithm detects to remove erroneous detections and add in any missed by the algorithm. The system tracks the origin of each minimum (i.e., whether it was added at the matched filtering stage, gap adjustment or manually by the user) and shown in the GUI. This ensures that artificial cycles resulting from the gap adjustment stage are distinguishable from filter- or user-defined minima, allowing for different assessment.

After confirming the count, the user specifies an anchor point to temporally constrain the annual boundaries, which could be the collection year, the radiocarbon bomb spike depth, or a U-Th point. The algorithm derives a median age model from 2,000 Monte Carlo realisations using piecewise cubic Hermite interpolating polynomial (PCHIP) interpolation, which it then uses to assign every datapoint in the depth-proxy dataset an age value. At this point there is the option to quantify algorithmic uncertainty and devise a confidence interval for the age model. Furthermore, CYCLIM can translate the proxy values onto a time-certain axis to convert age uncertainty into proxy uncertainty. Post-analysis, the user can export the minima locations with their origin (supporting further age modelling incorporating other chronological methods) and the age-converted dataset. Finally, the system facilitates the import and depth-age conversion of any additional proxies from the archive using the age model derived prior, provided it is within the same depth range. Once converted, the user can extract the age-converted data for further analysis.





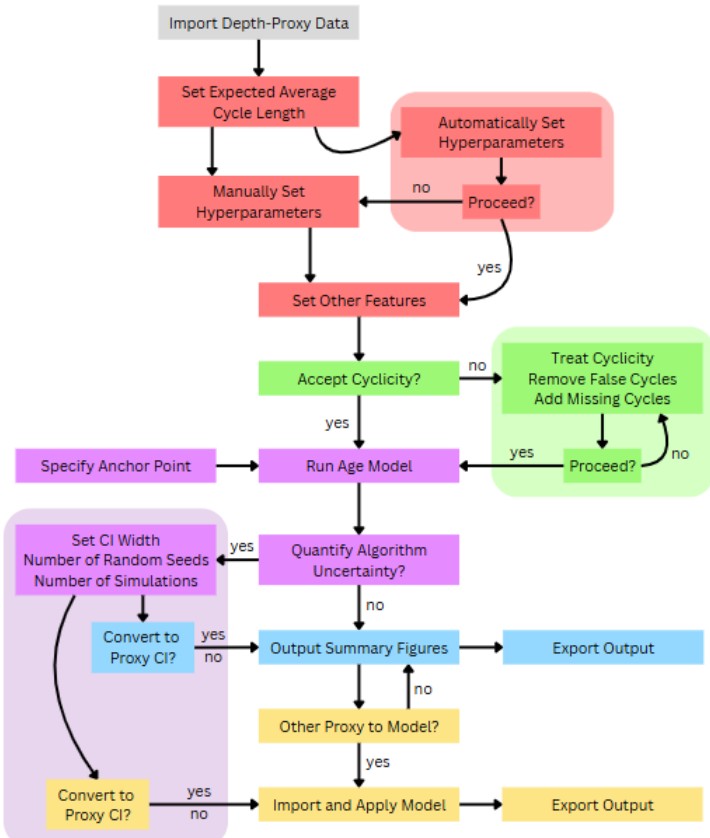

**Figure 1: A schematic overview of the CYCLIM workflow. Colours denote the step of the GUI in which each element occurs. Step 1 (grey): upload and confirm the depth-proxy data. Step 2 (red): hyperparameter set-up and minima detection. Step 3 (green): manual inspection of the output minima with the ability to add or remove entries. Step 4 (purple): run age model and compute uncertainties. Step 5 (blue): convert uncertainties and view final output. Step 6 (yellow): import and model additional depth-proxy datasets using the age model derived in Step 4.**

## 2.2 Cycle Detection

Cycles are extracted using a matched filtering approach designed to enhance the defined annual-scale cyclicity within the depth-proxy dataset (Fig. 2). A Gaussian-shaped template is constructed to approximate the expected waveform of the cycles. The user can manually specify the template width ($w$) or the CYCLIM algorithm will automatically set the template width to correspond to half the number of datapoints that comprise the average cycle length. The method uses a half-cycle length because matched filtering with a Gaussian template produces strong responses for segments that resemble either the template's standard orientation (peaks) or its inverse (troughs). This approach allows the filter to detect cycles without requiring knowledge of the full waveform shape, emphasising general pattern similarity and thereby making it robust to asymmetry or obscuration due to noise. Because the template centres around a single point, the window must be odd. If the calculation yields an even number, the algorithm expands the window by one. The Gaussian template is defined as a normalised function:




$$G(t) = \frac{1}{Z} \cdot \exp\left(-\frac{t^2}{2\sigma^2}\right),$$  (1)

where $t$ is an index vector centred on zero, $\sigma = \frac{w}{6}$, and $Z$ is a normalisation constant, computed as the sum of the unnormalised Gaussian values, that ensures the template has unit area. The choice of $\sigma$ ensures that $> 99\%$ of the Gaussian's area falls within the selected window.

The proxy signal data is then convolved with $G(t)$ to yield a matched filter output:

$$Y(t) = (y * G)(t),$$  (2)

where $*$ signifies convolution. The resulting signal $Y$ emphasises the regions of the original $y$ signal that resemble the template shape, thereby enhancing the cyclic component while reducing high-frequency noise.

Because the algorithm counts cycles using minima, it inverts the filtered signal before performing peak detection. Peaks in the inverted signal correspond to points that exceed their immediate neighbours and surpass a minimum prominence criterion.

This threshold suppresses false detections caused by remaining signal noise and allows the user to manually set the value or automatically assigns it a value equivalent to 2.5% of the range in $y$.

Following the initial detection stage, the algorithm can tune the found minima to a local minimum in the original dataset. If enabled, this feature uses the tuning window to search the nearby points in the raw signal of each minimum detected by the matched filter for a true minimum. In manual mode the user specifies the width of this tuning window, but in automatic mode

the algorithm sets the window to a quarter of the full cycle length, approximating a three-month period, whilst ensuring the window is odd by expansion where necessary. This ensures that final positions remain within the same season and correspond to inflection points in the original data.

To maintain accuracy and avoid undercounting, the algorithm includes an additional step to treat intervals where cyclicity is indistinguishable from background noise. If enabled, and no minimum appears within an interval exceeding the gap threshold

set by the user, the algorithm subdivides the section at regular intervals equal to the average cycle length and tunes each to localised minima. Although this step introduces artificial cycles, it ensures that the overall count does not deviate from the underlying age model due to missed cyclicity.




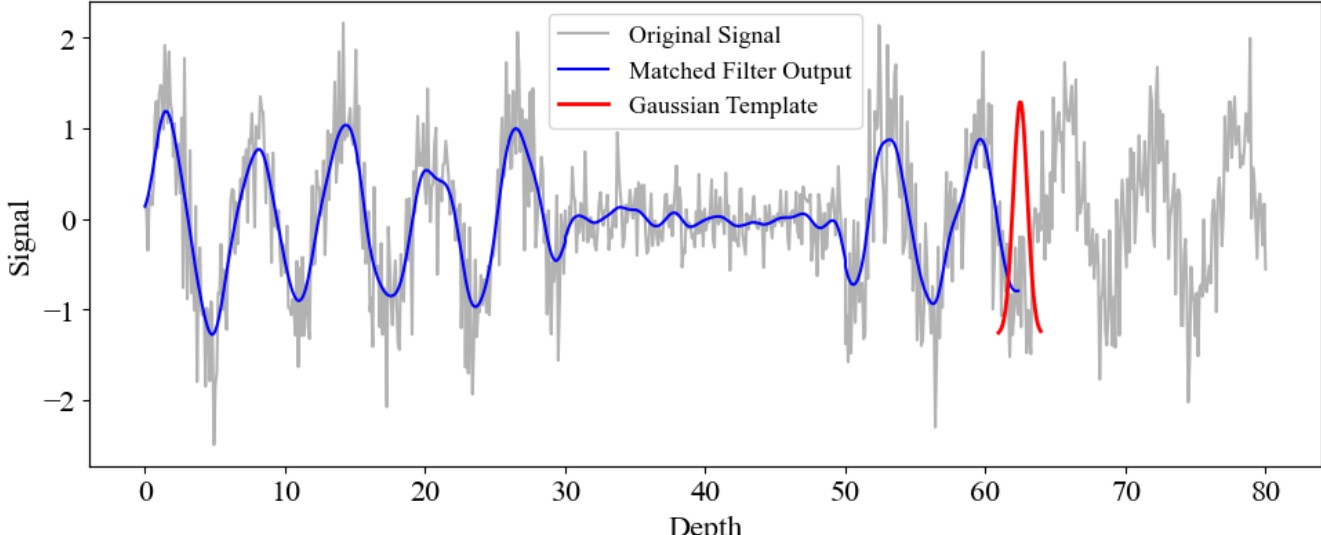

**Figure 2: A schematic overview of the matched filtering technique. A hypothetical dataset (grey) in the process of convolution (blue)**
**via the matched filtering of a Gaussian template set to half a cycle width (red; vertically expanded for clarity). Peak finding is then**
**performed on the matched filter output (blue).**

### 2.3 Quantifying Algorithm Uncertainty

To quantify algorithmic uncertainty as an approximation of cycle counting age uncertainty, CYCLIM adopts a noise-based
Monte Carlo approach. By perturbing the signal in a similar way to natural processes, we can model the algorithm's sensitivity
and yield estimates of variance between counts that reflect that arising from counting error.

First, CYCLIM stores the minima positions from the cycle detection stage, including those added by gap treatment, as a
reference set. For multiple random seeds, we then simulate random error by progressively overlaying the signal with Gaussian
white noise in 1% increments (from 1% to 100%). At each noise level, the same matched filter and gap treatment applied to
the reference set are re-run on each replicate. Each run's performance is then evaluated against the reference set using the F1
score, a harmonic mean of precision (here the fraction of output detections that are true) and recall (the fraction of reference
minima detected) based on a detection tolerance of a quarter cycle length. The analysis employs the F1 because it includes
both over- and undercounting as systematic error, making it a robust metric of signal recovery fidelity under increasing noise.
The mean 95% confidence interval for the F1 scores across all seeds at each noise level is computed, producing a performance
curve (Fig. 3a).
To identify the critical noise threshold whereby algorithm performance begins to degrade disproportionately to the increase in
noise, the first and second derivatives of the F1 score curve with respect to noise level are calculated (Fig. 3b and c). The first
minimum in the smoothed second derivative provides an estimate of this noise level, marking the lowest level of noise capable
of inducing a meaningful loss in detection robustness.



The tipping level represents the boundary between algorithm robustness and failure, where detection accuracy becomes increasingly sensitive to small changes in noise level showing the algorithm's operational limit. Modelling the algorithmic response to lower (higher) levels of noise could underestimate (overestimate) uncertainty and so simulating at the tipping noise level avoids misrepresenting uncertainty. By using this level, the internal biases of the matched filter arising from hyperparameter choice become the salient driver of variability across random seeds. It thus exposes how stable the algorithm's output is to noise, replicating signal ambiguity due to discrete sampling. Therefore, it mimics the plausible levels of epistemic uncertainty of manual counting by perturbing the signal enough to induce detection error but without causing irreparable signal damage. Additionally, because the threshold is dynamic it is less sensitive and can produce accurate estimates for a wider range of input signals.

Replicates of distorted signals are constructed by combining the signal with the random seeds at this threshold noise level. The resulting ensemble is passed through the cycle detection algorithm using the same inputs, and the locations of each run's detections are stored, providing a record of internal model variance. The ensemble of age models is then linearly interpolated onto a common depth axis provided by the reference set and percentile-based confidence intervals for the variance in number of detections per reference set entry is calculated (Fig. 3d). These detection uncertainties are then mapped onto the final age model (i.e., that post-adjustment), giving a realistic estimate of counting error. Additionally, the age model uncertainties can be translated into time-certain proxy uncertainties via ensemble-based mapping of proxy values onto the time domain.





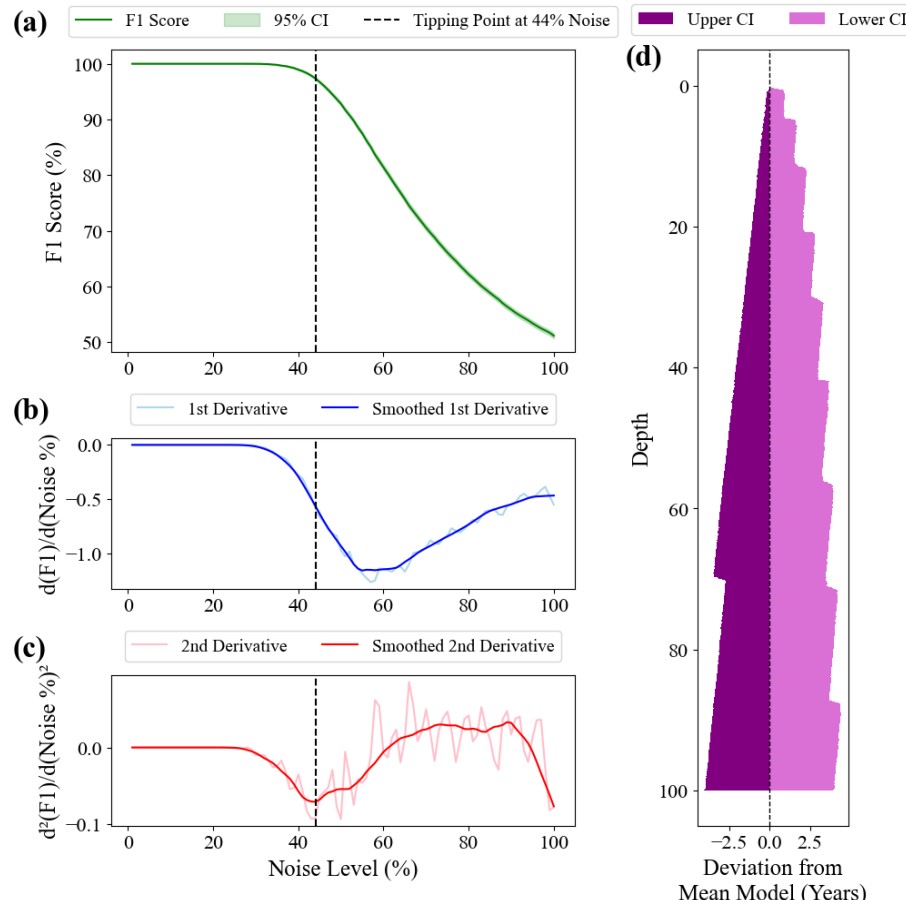

**Figure 3: A schematic overview of algorithmic uncertainty calculation using a dataset of 100 sine waves. (a) The algorithm's ability to reproduce the same list of minima under increasing noise levels. (b) and (c) the first and second derivatives respectively of the output shown in (a). (d) the variance in number of minima detected with depth under the tipping point level of noise. The 95% confidence interval for 100 random seeds is shown relative to the mean value. Note that because the synthetic data is a perfect sine wave, noise is more likely to lead to more detections rather than obscure existing cycles hence the imbalance, and that the stepped nature is the product of discrete random sampling.**

## 3 Example Use of CYCLIM

To test CYCLIM's performance, we show its application to three previously published palaeoclimate records. All three show annual scale cyclicity but with varying degrees of clarity and consistency. For each example the automatically generated template size hyperparameters were used, but the minimum prominence criterion was sometimes changed due to record extrema effecting the automatic estimate. Additionally, these tests were run without knowledge of the published cycle count. The mean cycle length was extracted from the depth-proxy information and the manually tuning was performed independently, without comparison to the original paper's age model.



## 3.1 Records

### 3.1.1 Houtman Abrolhos Coral

As a first-pass test of CYCLIM's ability to detect cycles we use the coral $\delta^{18}O$ record from the Houtman Abrolhos Islands off west Australia presented in Kuhnert et al. (1999), which exhibits very prominent annual cyclicity with a very consistent annual growth rate and low noise level (Fig. 3). The core was sampled from a living colony of *Porites lutea* in May 1994 and is 300 cm in length. The $\delta^{18}O$ record likely captures past temperatures, with high-frequency fluctuations reflecting annual cyclicity.

Ages were thus determined by counting this cyclicity relative to the date of collection. The original chronology spans 200 years (1794-1994 CE). Whereas the annual variability is clear throughout almost all the record, the authors note that the clarity of the $\delta^{18}O$ cycles weakens near 1853 and 1869. Estimated age errors are 1 year per century (Kuhnert et al., 1999).

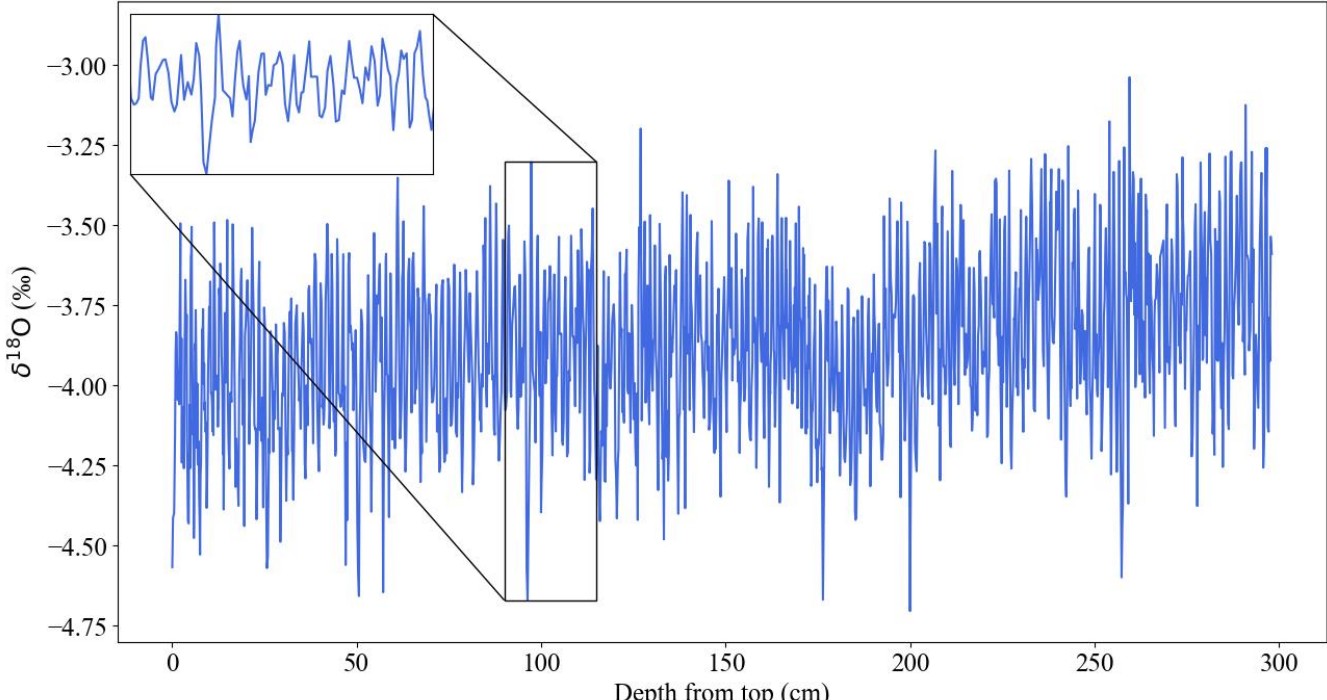

**Figure 4: The depth-proxy Houtman Abrolhos Coral $\delta^{18}O$ record (blue; Kuhnert et al. (1999)). The inset shows an example interval**
**of annual scale cyclicity.**

### 3.1.2 Stalagmite C09-2

Next, we use the stalagmite record presented in Warken et al. (2018) which exhibits a variable growth rate and more pronounced noise, to test the algorithm's ability to cope with inconsistent waveform characteristics. The stalagmite was collected from a cave in southwest Romania in 2009 and is 70 cm tall. Trace element variability exhibits pronounced cyclicity,

likely driven by strong seasonal differences in calcite precipitation rates (Warken et al., 2018). Mg/Ca, Ba/Ca, and Sr/Ca all show annual cyclicity, but only Sr/Ca is used here for cycle counting. The uppermost 1 mm was lost during sample preparation



and the cyclicity is lost after 47 mm due to an abrupt decrease in growth rate. The original cycle count found cycle of lengths 0.05 to 0.6 mm with a mean length of 0.21 mm (Warken et al., 2018). This means that cycle lengths can vary between 4 and 48.5 datapoints presenting a challenge for the algorithm's set template width. The 1955 CE bomb spike depth anchors the

chronology and was found to be at 3.2 mm depth. The original layer counted chronology spans 214 years (1759-1973 CE). Suggested cycle counting errors are ±3 years (Warken et al., 2018).

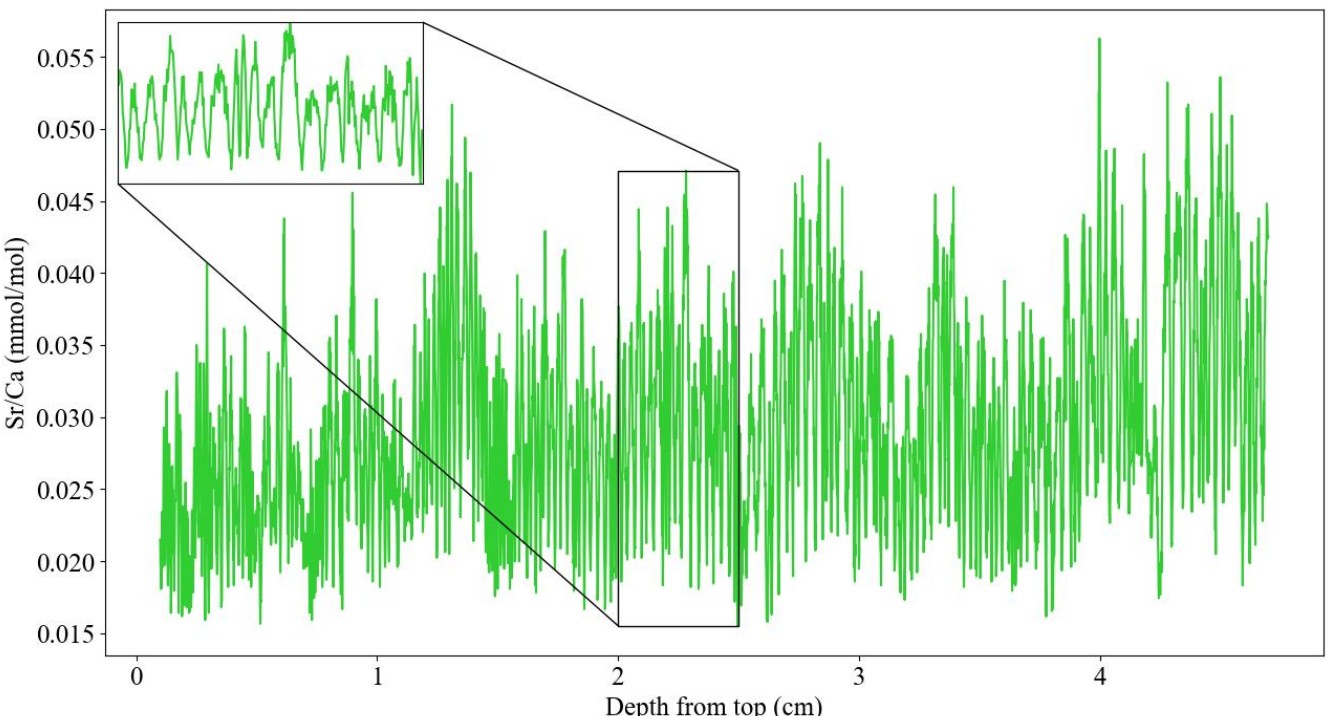

**Figure 5: The depth-proxy C09-2 stalagmite Sr/Ca record (green; Warken et al. (2018)). The inset shows an example interval of annual scale cyclicity.**

**3.1.3 Stalagmite BER-SWI-13**

As a final example, we use a speleothem-derived record spanning 564 years (1449-2013 CE) with overall well-developed geochemical cyclicity but with intervals of more ambiguous cyclicity presented in Forman et al. (2025) to test CYCLIM's ability to respond to record gaps (Fig. 4). This stalagmite was collected in 2013 from Leamington Cave in Bermuda and is 19.2 cm tall. The chronology uses radiocarbon dates modelled using the technique presented in Lechleitner et al. (2016) and

developed further in Fohlmeister and Lechleitner (2019) to obtain an 'average' chronology, which suggests an overall average growth rate of ~0.345 mm yr$^{-1}$. The 1955 CE bomb spike was determined to be at a depth of ~8 mm. A Mg dataset with annual cycles was then used to refine the radiocarbon-based model.

The stalagmite's fast response to rainfall and site's clear seasonal temperature and wind speed variability means that the proxy exhibits annual scale cyclicity but with a low signal-to-noise ratio. Magnesium cycle counts were performed by inspection.



The top ~5.4 mm exhibits pronounced cyclicity loss possibly arising from anthropogenic hydrological changes. Short intervals throughout the record similarly see the cyclicity break down. The original chronology thus models these sections to the radiocarbon average cycle length to maintain chronological accuracy.

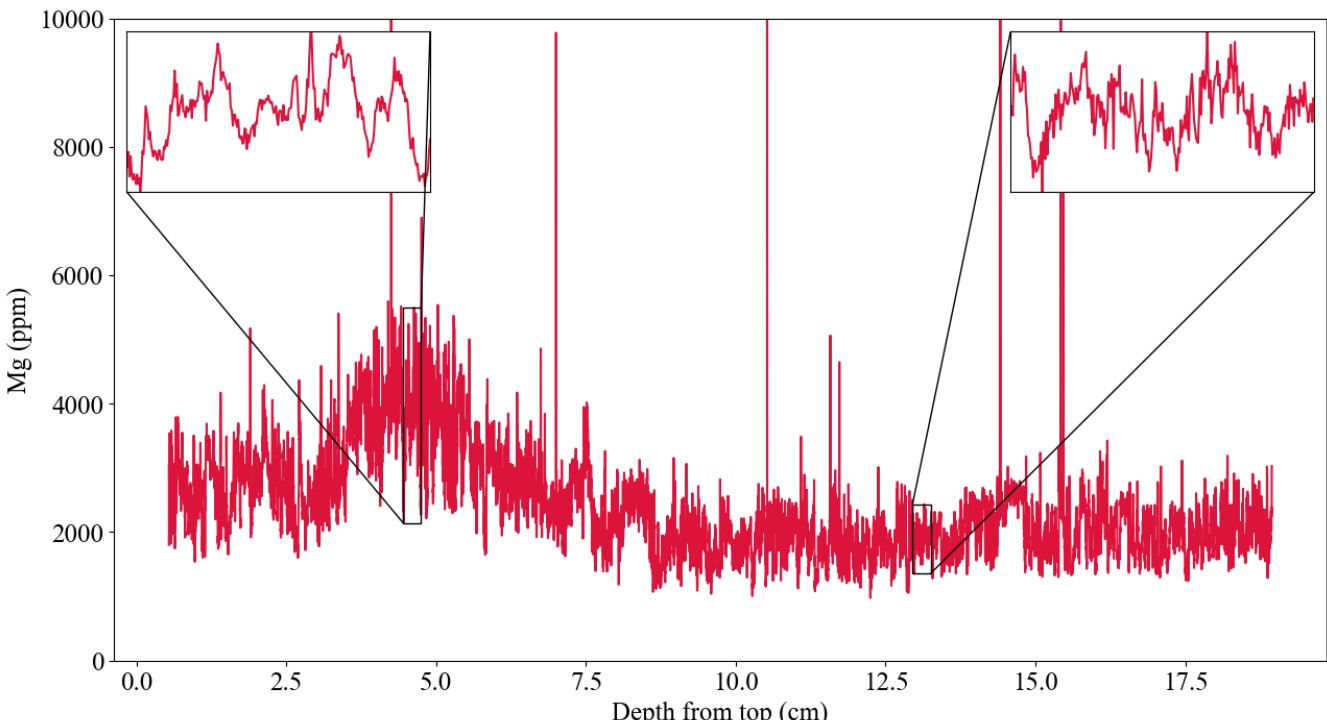

**Figure 6: The depth-proxy BER-SWI-13 stalagmite Mg record (red; Forman et al. (2025)). The left inset shows an example interval**
**of annual scale cyclicity and the right shows an example of less obvious/absent cyclicity mid-record. Note that the y axis extent is adjusted so that extrema do not reduce figure legibility.**

## 3.2 Results

### 3.2.1 Houtman Abrolhos Coral

The coral record exhibits very clear cyclicity with an average distance of ~1.58 mm, as approximated by spectral analysis. The
average distance is approximately 8.11 datapoints and so the algorithm prompts the use of template and tuning windows of 5 and 3 datapoints, respectively. Using the range in y values the starting minimum prominence value is 0.04‰.

The automatic age model achieves a good match with that published by Kuhnert et al. (1999), albeit with clear signs of overcounting (Fig. 7a and c). The overcount increases with depth to a maximum of 8.05 years (mean absolute error = 4.56 years, 2.3% of the target's temporal range), suggesting that the minimum prominence value is too low. Raising it to 0.08‰
confirms this, after which the temporal mismatch throughout the record becomes negligible (mean absolute error = 0.51). Correcting the overcount within the cycle tuning stage is equally powerful to changing the minimum prominence but also does not rely on backcalculating hyperparameters (Fig 7b and c). Visual inspection found 7 false cycles and no missing cycle,





meaning that the automatic output found all the cycle boundaries and achieved an F1 score of 0.98. Post-correction the output's discrepancies (maximum of 1.55 years, mean absolute error = 0.54 years) simply reflect the reproducibility uncertainty of

cycle counting by inspection. Uncertainty modelling using 1,000 random seeds identified a 5% noise tipping level, prompting 10,000 simulations at this ratio to estimate age errors. The algorithm's 95% confidence interval agrees with the published uncertainty of ±1 year per century with an uncertainty of ±2 years at maximum depth (Fig. 8).

**Figure 7: The CYCLIM output for the Houtman Abrolhos Coral record. (a) The automatic age-converted output (light blue) shown**
**with the original record (black) presented in Kuhnert et al. (1999). (b) The output (dark blue) after manual cycle tuning inspection.**
**(c) The temporal offset needed to restore the original record at a given point in the automatic (light blue) and manual (dark blue)**
**model output.**





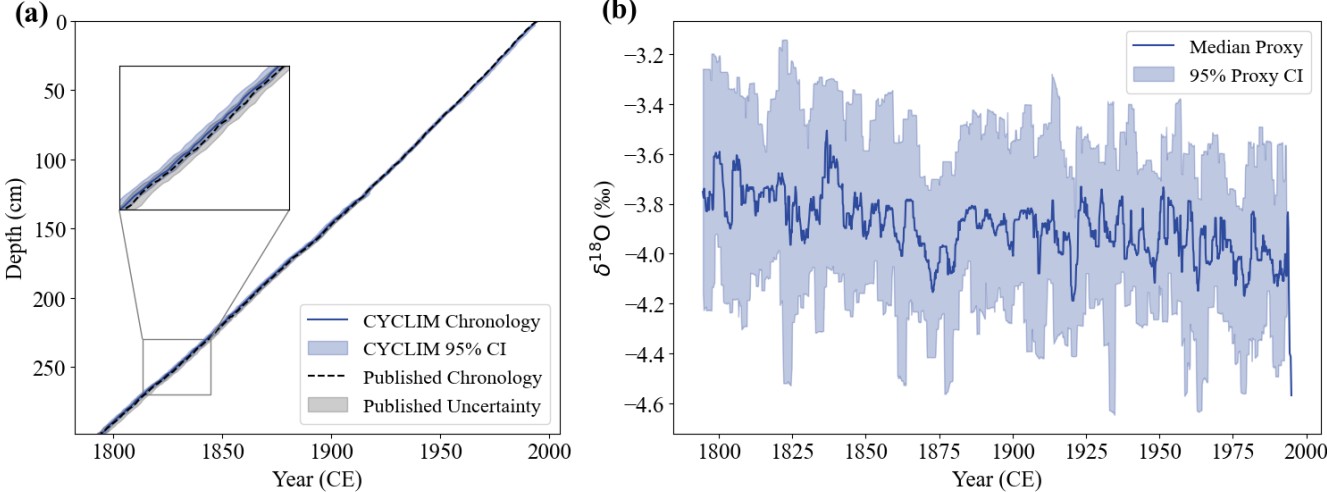

**Figure 8: The CYCLIM age model and age-certain output for the Houtman Abrolhos Coral record. (a) The manual CYCLIM age model (blue) shown with the published age model (black) presented in Kuhnert et al. (1999), both with their respective uncertainties. (b) The manual CYCLIM output record with age-proxy uncertainty translation on a time-certain axis.**

### 3.2.2 Stalagmite C09-2

The C09-2 stalagmite record has a more variable cycle length and contains more noise, presenting a test of the algorithm's versatility (Warken et al., 2018). Spectral analysis suggests the average annual cycle length is approximately 0.292 mm, which translates to 23.6 datapoints. The automatic hyperparameter choices thus become 11 and 5 points for the template width and tuning window, respectively. Additionally, the 2.5% minimum prominence corresponds to 0.001 mmol/mol.

The automatic output exhibits some agreement with the record published in Warken et al. (2018), however clear undercount exists in both directions of the chronological anchor point (Fig. 9a and c). This leads the record to have an accumulating mismatch which rises to a maximum of 8.23 years by the end of the record at 207 years counted (mean absolute error = 6.08 years, 2.8% of the targets temporal range). To correct the undercount, a minimum prominence a third of the automatic value (i.e., 0.0003 mmol/mol) was tried, yielding a stronger agreement (mean absolute error = 1.57 years). However, instead we use the initial values and correct the output in the tuning stage (Fig 9b and c). Manual inspection found no erroneous cycles and resulted in the addition of 12 cycles, meaning the automatic output found 94.4% of the cycles and achieved an F1 score of 0.97. The tuned record has a maximum absolute mismatch of 1.77 years (mean absolute error = 0.41 years) showing a strong agreement with the published record. Uncertainty quantification used 1,000 random seeds and 10,000 simulations at the determined 5% noise tipping level. Uncertainties broadly follow those presented in the Warken et al. (2018), albeit slightly lower (Fig. 10). At maximum depth, the 95% algorithmic uncertainty is approximately ±2 years compared to the published ±3 years.







Figure 9: The CYCLIM output for the stalagmite C09-2 record. (a) The automatic age-converted output (light green) shown with the original record (black) presented in Warken et al. (2018). (b) The output (dark green) after manual cycle tuning inspection. (c) The temporal offset needed to restore the original record at a given point in the automatic (light green) and manual (dark green) model output.



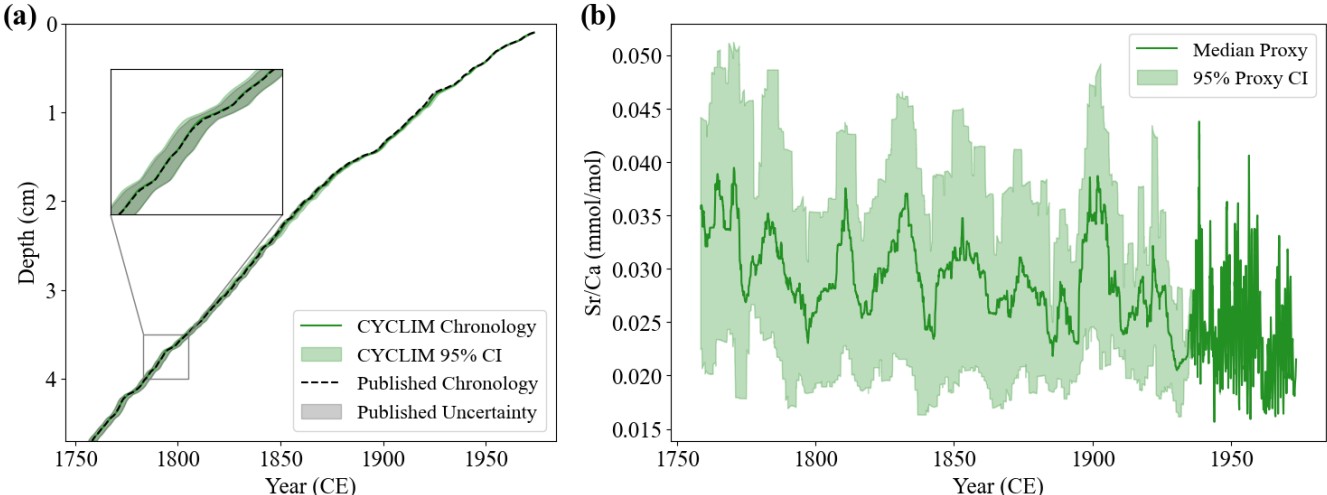

**Figure 10: The CYCLIM age model and age-certain output for the stalagmite C09-2 record. (a) The manual CYCLIM age model (green) shown with the published age model (black) presented in Warken et al. (2018), both with their respective uncertainties. (b) The manual CYCLIM output record with age-proxy uncertainty translation on a time-certain axis.**

### 3.2.3 Stalagmite BER-SWI-13

This final dataset assesses the algorithm's ability to respond to gaps in the record. Radiocarbon modelling of stalagmite BER-SWI-13's growth suggests a mean cycle length of approximately 0.345 mm which corresponds to ~33.6 datapoints. The template width and tuning window therefore become 17 and 9 points, respectively. The tool prompts the use of 503 ppm for a minimum prominence, but this is heavily skewed due to extreme outliers in the data and so it is manually adjusted to 80 ppm. Additionally, gap treatment was applied when no cycles were found within 1.035 mm (i.e., three average cycle lengths). The

top ~5.4 mm is subjected to complete cyclicity loss and so the original paper models this section so that the top matches the collection year, but due to the substantial chronological uncertainty over this interval, this section is left out of the input.

The generated age model shows broad agreement with the original record achieving an overall mean absolute error of 3.88 years (0.8% of the target's temporal range) (Fig. 11a and c). At ages younger than 1800 CE there is strong agreement between the two age models, but a serial undercount emerges thereafter reaching a maximum offset of 12.56 years, before being

counteracted by a subsequent overcount. The trend in mismatches suggests the algorithm is subdividing cyclicity gaps that are different to those in the original, because the original ambiguous sections were determined by inspection and not by a threshold. Tuning found 33 false and 30 missing cycle boundaries, meaning that the automatic output found 94.1% of the cycles in the final output and an F1 score of 0.94. After manual correction of the output (Fig. 11b and c), the age model follows the original more closely with a maximum absolute deviation of 3.63 years (mean absolute error = 1.18 years). Given the record extends

over five centuries and contains gaps, this remaining degree of mismatch is within reproducibility error. Uncertainty modelling with 1,000 random seeds and 10,000 simulations found a 13% tipping noise threshold, which led to an uncertainty of ±13.5 years at maximum depth (Fig. 12).





**Figure 11: The CYCLIM output for the stalagmite BER-SWI-13 record. (a) The automatic age-converted output (pink) shown with the original record (black) presented in Forman et al. (2025). (b) The output (red) after manual cycle tuning inspection. (c) The temporal offset needed to restore the original record at a given point in the automatic (pink) and manual (red) model output.**



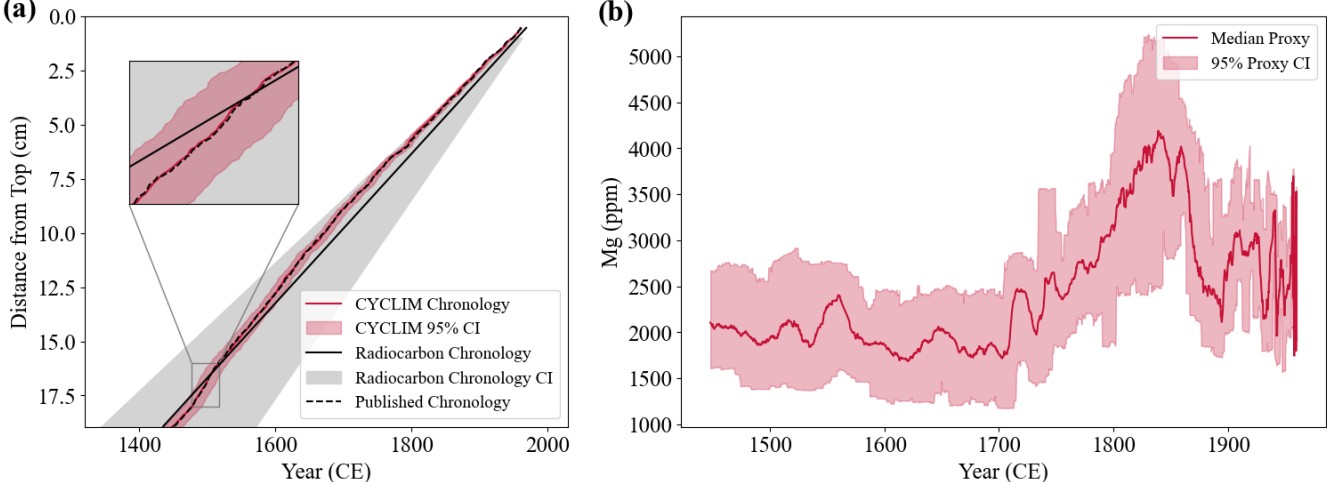

**Figure 12: The CYCLIM age model and age-certain output for the stalagmite BER-SWI-13 record. (a) The manual CYCLIM age model (red) with 95% confidence interval, shown with the published age model (black) and the radiocarbon carbon chronology with 2σ uncertainties presented in Forman et al. (2025). (b) The manual CYCLIM output record with age-proxy uncertainty translation on a time-certain axis.**

## 4 Discussion

The CYCLIM algorithm extracts cyclicity information from three examples both accurately and quickly via an automated matched filtering technique, achieving close approximations of the original chronologies. Across the three examples the temporal offsets ranged from -8.05 to 12.56 years with an average mean absolute error of 4.84 years before correction. Additionally, there was an average 3.26% erroneous detection error and 3.81% missed detection error, meaning that cycle counting within CYCLIM was ~14.1 times faster across the three examples. Whereas its accuracy does depend on the choice of hyperparameters, only two require estimation. Because other methods derive the average cycle length estimate, this is a fixed value leaving only the minimum prominence at the discretion of the user to change. The automatic minimum prominence value is sensitive to outliers in the data, but users can easily override this with a better estimation from inspection of the output. Thus, the matched filtering approach is robust and can greatly speed up the cycle counting process by providing a first-pass count for the user to refine.

Post-analysis users should check and refine the automatic output in the cycle tuning stage to mitigate issues such as: (1) overcounting resulting from gap treatment; (2) undercounting due to low signal-to-noise; (3) a change in average cycle width hindering hyperparameter effectiveness; and (4) erroneous hyperparameter choices. The results from this stage follow the original chronologies very closely and further running could yield a perfect match. However, to ensure a fair test, tuning was conducted independently and without back-calculation of cycle boundaries using the published chronology. Thus, whereas the outputs are not perfect reflections of the originals, they are within the expected counting errors, hence demonstrating the efficacy of the CYCLIM tool.



The two-stage semi-automated approach generates high-precision cycle counts within the errors expected of count reproducibility and could be applied to a wide variety of archives, provided the assumptions hold. CYCLIM could have several potential uses besides simply generating a chronology, for example: (1) reporting an objective cycle count (i.e., the algorithm output) before correction; (2) easily running multiple counts to test reproducibility; and (3) reporting an objective count before then running multiple corrections to develop an average age model.

**5 Conclusions**

Cycle counting is a powerful tool capable of deriving chronologies of sub-annual resolution. However, the process is often time consuming, and the output can vary between counts. Speeding up this process would therefore allow researchers to perform more counts, which could lead to a better understanding of a record's counting error. Various automated methods exist but many users prefer manual counting for its the simplicity, versatility, and transparency.

Here we present a new semi-automated python-based application (CYCLIM) for deriving chronologies from annual-scale cycle counting. The algorithm performs an initial cycle count using matched filtering, after which the user may inspect and refine the output at their discretion by removing false detections and/or adding in missed cycles. Based on the testing presented here, CYCLIM generates quick and accurate automatic counts. The tuning stage provides the user the space to correct the output where needed within a user-friendly GUI and export the results for further analysis. One application of CYCLIM is to

produce an automated count and run multiple user corrections to assess the counting errors associated with reproducibility. CYCLIM is freely available via the Zenodo repository (https://doi.org/10.5281/zenodo.16651941).

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
