# Peer review of "CYCLIM: a semi-automated cycle counting tool for palaeoclimate reconstruction"

_EGUsphere, 2025_

## Author Comment (AC1)

**Response to Reviewer comments, Forman and Baldini, submitted to *Climate of the Past**

We thank the reviewers for their helpful suggestions and appreciate the time and effort that went into reviewing the previous version of the manuscript. In this response letter we address all the comments. All replies are in blue and relevant passages from the revised manuscript are in red.

**Reviewer #1 comments:**

Comment #1: In this paper the authors present a code, which is able to automatically count isotope-geochemical growth bands. The aim is to provide a fast method to perform the annual layer counting task to establish a precise chronology. While there have been some approaches of automated layer counted algorithms available in the field, the new aspect of their approach is to allow the user a refinement after the automated counting procedure. However, even the counting algorithm alone seems to perform quite well, when judging the results of the three provided examples, each of different complexity. With the authors approach, the user can perform the counting-task much faster than when counting all cycles alone.

The manuscript is well written and structured. Especially, the introduction appears to be very nice to me. I also liked the section with the examples and the short discussion. What could be somewhat improved is the methods part, which is the most important section in the manuscript. I miss a bit more details on the model parameters and their influence on the layer counting. Please find more details on my – what I would call – moderate suggestions listed below. Pending on improvements with respect to the suggestions, I suggest to consider this manuscript for publication in CP. To my opinion the manuscript deserves to be published as I can imagine, that many researchers can and will make use of this approach.

Response: We thank the reviewer for their positive assessment of the paper and for highlighting the lack of transparency with the methods, which we address in the comments below.

**Comment #2**: L14 and L311-312: The term '14.1 times faster' is very specific. And unfortunately, it is not explained in the text, how this number is calculated. At the moment, I doubt this number. Especially, that this number is always – for all records no matter how difficult or long they are – 14.1. I suggest to change this, to something broader, something like: 'one order of magnitude faster'.

Response: The 14.1 statistic came the user only having to count/correct 7.1% cycles (3.3% false positive and 3.8% false negative). However, because it does not take into

account the actual timings, we have changed the sentence in line with the reviewer's suggestion to make this more accurate.

**Comment #3**: L75-78: This part is about the parameters. Unfortunately, this part is relatively poor, to my opinion. It is not described at all, how the parameters are derived by the code. There is also no real description about what the parameters are and what they are steering or how they influence the result. In the results section, where the approach is tested against already published data, there is some text, which helps to at least adumbrate how some of the parameters are derived. I suggest to at least add a table where the necessary parameters are listed, where it is shortly explained how they are determined or what are typical values. Maybe it is also helpful to leave some sentences in the text or table, which describe them in more detail.

Interesting for the reader may also an answer to the question, what the choice of the mean cycle length has an influence on the result – at least for the automated part. – I guess the total influence on the result is minor, as the manual part can change the result in an arbitrary way.

Response: We have made the hyperparameter methods clearer, stating what they are, their purpose and the output's sensitivity to their value. The changes include additional text and a table (Table 1).

Assuming the average cycle length is reasonably constrained, the sensitivity to its value is relatively minor. As shown in the Warken et al. 2018 example and Fig. 10a, the algorithm still returns accurate cycle boundaries even if the estimate is slightly off the "true" mean. However, if the record resolution less than ~8 points per cycle, changing the estimate could lead to dramatic changes in template size. The strength of the signal-to-noise ratio also determines the sensitivity of the output to a change in template size.

The following text was added: (Lines 137-145)

The detection methodology thus uses four hyperparameters: (1) the average cycle length; (2) the template width; (3) the minimum prominence; and (4) the tuning window (Table 1). In custom mode, the user must specify values for all of the hyperparameters, whereas in default mode, only an estimate for the average cycle length is required. However, because this estimate is used to determine the value of the other hyperparameters it needs to be a fair approximation. Provided the estimate is robust, the sensitivity of the output to its exact value depends on the record's resolution and noise level. The sensitivity of the output to the choice of template width and minimum prominence is computed within the algorithm. The GUI then plots the similarity of the detected cycle boundaries relative to the current configuration as the template width and minimum prominence are varied. The template width is modified by ±30% and the

minimum prominence value is converted to a percentage to the range in y and adjusted by ±5%.

and Table 1, which reads: (Lines 151-152)

| Hyperparameter          | Description                                                                | Default Value                                                          | Purpose                                                                                                                         | Effect on
Output                                                                                 |
|-------------------------|----------------------------------------------------------------------------|------------------------------------------------------------------------|---------------------------------------------------------------------------------------------------------------------------------|-----------------------------------------------------------------------------------------------------|
| Average Cycle
Length | Estimated length (in depth units) of the typical cycle in the proxy signal | User-defined                                                           | Serves as a basis for deriving the template width and tuning window. It is also used to divide up sections during gap treatment | Directly influences default values of the other hyperparameters and the length of gap-filled cycles |
| Template Width          | Width of the
Gaussian
template used
for matched
filtering      | Half the number of datapoints corresponding to an average cycle length | Determines
how much of
the signal is
used in each
convolution
window                                             | The wider the template the smoother the output will be                                              |
| Minimum
Prominence   | Minimum prominence required to accept a detected cycle                     | 2.5% of the
proxy signal's
central 95%
range                  | Filters out
false
detections
caused noise                                                                              | The greater the value the higher the peak must be                                                   |
| Tuning Window           | Local window
to refine
detected cycle
positions                   | One-quarter of
the datapoints
in an average
cycle length      | Tunes the detections by aligning them to match a local minimum in the original proxy signal                                     | The higher the value the wider the scope for searching                                              |

**Comment #4**: L78-79: What happens with gaps due to cuts of the samples as they might occur during sample preparation? Are they also covered by this?

Response: Gap treatment would divide gaps incurred during sample preparation by the average cycle length. This can be used to keep the age model in line with what is expected by the archive's growth rate. Alternatively, the user could manually insert a given number of cycle boundaries in this gap or upload the record in two sections running the algorithm on each, provided there is an anchor point available in both. This is now mentioned in text: (Lines 132-133)

To maintain accuracy and avoid undercounting, the algorithm includes an additional step to treat missing sections of the record and/or intervals where cyclicity is indistinguishable from background noise.

Comment #5: L80-83: This is a very helpful tracking procedure. Very nice idea.

Response: We thank the reviewer for their assessment.

**Comment #6**: L84-85: You are counting possible types of anchor-points here – i.e. only one of each type. However, I can imagine, that especially for U-Th ages, there could exist more than one dated depth over the counted interval. Is CYCLIM able to cope with that as well? Including all (possible) U-Th dated depths (if available) in placing the counted interval could really help to pinpoint the chronology.

Response: We thank the reviewer for this suggestion and have included this as a feature within CYCLIM. Alongside an anchor point (no uncertainties) the user can now upload a list of tie points with errors. These are displayed within the GUI so the output can be verified. Additionally, there is an option to reject Monte Carlo realisations during uncertainty quantification that plot outside the tie point errors. While the examples do not have a full usable radiometric age models to showcase this feature, it is applied on stalagmite C09-2 with the inclusion of one U-Th date alongside the anchor. This added feature is discussed in text: (Lines 86-87)

There is also the option to upload a full list of chronological tie points for comparison and specify the sub-annual timing of the minima.

and (Lines 189-190)

Also, if additional chronological tie points are uploaded there is the option to use them to constrain the realisations by skipping age models that plot outside of their uncertainties.

and (Lines 308-309)

Simulations were constrained by the U-Th date uncertainties and led to the rejection of 232 realisations.

**Comment #7**: L85-86: "The algorithm derives a median age model from 2,000 Monte Carlo realisations using piecewise cubic Hermite interpolating polynomial (PCHIP) interpolation, which ..." Can you please elaborate a bit more on this? I don't understand why you are needing interpolation here. And what kind of Monte Carlo realisations? What is varying?

Response: We thank the reviewer for finding this error. Interpolation is performed so that ages can be approximated between cycle boundaries, but the reference to Monte Carlo realisations pertains to the uncertainty analysis and so should not be mentioned here. The text has been updated to read: (Lines 87-89)

CYCLIM then uses piecewise cubic Hermite interpolating polynomial (PCHIP) interpolation to assign an age value to every datapoint in the depth-proxy dataset by estimating ages between the detected cycle boundaries.

Comment #8: L 86: "uses" à "used".

Response: Corrected.

**Comment #9**: L88-L89: "Furthermore, CYCLIM can translate the proxy values onto a time-certain axis to convert age uncertainty into proxy uncertainty." This is not a specific comment to this paper, and you are free to ignore, if you like, but maybe you can help me out here.

I know this concept has already been proposed in earlier studies (e.g., Breitenbach et al., 2012). However, I, personally, do not really understand this concept. It appears not to be meaningful to me. I always think about this the following way: Only as a signal cannot be put perfectly in time, the magnitude of an event is not smaller than measured. Or more extreme, only as a clearly pronounced event in the measured proxy cannot be dated at all, it does not mean it is not there at all, as this approach would tend to suggest. At least this is my argument for not agreeing with this approach. But this is only my opinion about this issue. Maybe there are arguments in favour of this procedure, I am not aware of.

So, my question to this sentence would be, if this feature can be deselected by the user?

Response: We appreciate the concern with this feature and agree that the approach has limitations. While some of the resolution of the record is lost, it enables the direct comparison of multiple records and/or statistical analysis without needing to consider age uncertainty which could be useful for some users. This feature is optional and, in line with this comment, we have adapted the following sentences to read: (Lines 90-91)

Furthermore, users can choose to enable the translation of the proxy values onto a time-certain axis, thereby converting age uncertainty into proxy uncertainty.

and (Lines 196-197)

Finally, the age model uncertainties can optionally be translated into time-certain proxy uncertainties via ensemble-based mapping of proxy values onto the time domain.

**Comment #10**: L103-106: Could you elaborate a bit more on the choice of the width, w? I think this would help the reader, to set their own w if they prefer to do so. What is the

impact of a change in w? Can you perform a short sensitivity analysis? Maybe with one of your example data sets (or all).

Per default CYCLIM is using the half average annual cycle length. However, growth layers can change quite strongly throughout the record. What would happen to phases, where the cycles are much shorter than the average. And what would happen to phases, where there is very rapid growth? Does such a behavior result in an under- or overcounting. I think, this would be very interesting to the reader. At least to me, it is.

Related to this, it might be worth in another, future version of CYCLIM to make the template width, w, depth adaptive by using a wavelet. In case the over- and undercounting in periods with low and high growth rates is an issue, this method could improve the counting performance, when changes from fast to slow growth occur. Just an idea.

Response: By using a half cycle length, the template becomes more robust and flexible to the changes of a proxy signal. If the template was a full wave cycle, then asymmetric cycles with would be more difficult to identify. Additionally, with this method the matched filter output still identifies the annual signal even when the growth rate changes slightly. The text has been adapted to better explain the hyperparameter (please see extract below and Comment #3): (Lines 105-112)

A Gaussian-shaped template is constructed to approximate the expected waveform of the cycles. The user can manually specify the template width (w) or the CYCLIM algorithm will automatically set the template width to correspond to half the number of datapoints that comprise the average cycle length, which is specified by the user. The method uses a half-cycle length because matched filtering with a Gaussian template produces strong responses for segments that resemble either the template's standard orientation (peaks) or its inverse (troughs). This approach allows the filter to detect cycles without requiring knowledge of the full waveform shape, emphasising general pattern similarity and thereby making it robust to asymmetry or obscuration due to noise. Because the template centres around a single point, the window must be odd. If the calculation yields an even number, the algorithm expands the window by one.

Regarding the template width's sensitivity to changes in growth rate, it is primarily dependent on the signal-to-noise ratio as the template is only used to smooth the record. Therefore, varying growth rate can be accommodated by the template width provided the signal is clear and the hyperparameter is set to the half the average cycle length. This is now visualised in Figures 8a, 10a, and 12a (example below), which show the extracted cycle lengths of the automatic vs published age models. All three example show the algorithm can still accurately detect cycles when there is significant variance in length. Also, as long as its near constant any growth rate changes can be accounted for in the manual tuning stage.

Figure 12: The CYCLIM age models and manually tuned output for the stalagmite BER-SWI-13 record. (a) Distribution of annual cycle lengths for the automatic (pink) and published (grey) chronologies. The vertical dashed line shows the average cycle length hyperparameter estimate. (b) The CYCLIM age models (automatic – pink; manual – red), shown with the published age model (black) and the radiocarbon carbon chronology with 2σ uncertainties presented in Forman et al. (2025). (c) The manual CYCLIM output record (red) and chronological mismatch with the published record (purple). Mismatches are averaged by decade.

To help users better understand the output's sensitivity to the chosen hyperparameter configuration, we have added a visualisation to the GUI showing how well moderately different hyperparameter choices return the current list of detected minima. This plot (example shown below) plots the similarity of outputs with moderately different template width and minimum prominence values to the current output using a heatmap of F1 scores. The template width is varied by ±30% and the minimum prominence is converted to a percentage of the range in the proxy signal and adjusted by ±5%, allowing for a large coverage of possible combinations. However, in some cases these values may become inaccurate, or the current combination could be very robust but sensitive to change. Thus, the sensitivity should be assessed with caution. A copy of these plots for each of the three example is now available in the manuscript in Figures 7, 9, and 11.

The reviewer suggests a good idea to use a wavelet to track the cyclicity signal, and thus the template would be adaptive to changes in growth rate. However, because the template width is inherently flexible (as shown in the sensitivity analysis) this approach may introduce too much complexity. It would require the code being able to find the correct wavelength amongst potentially many others and would assume that there is a cyclicity to find. Thus, it would need to be constrained and risks a more black-box methodology. For this reason, we have kept the fixed template width method to make the output easier to understand.

Comment #11: L119-221: "minimum prominence criterion" Please elaborate more on this. What is this? How did you (or CYCLIM) is defining this? Have you tested the impact of this criterion? Is this the same as what you are later, in sections 3.2.1 to 3.2.3, are setting to 0.04 permil (Houtman Abrolhos Coral), 2.5% (C09-2), 503 ppm (BER-SWI-13). You may be able to test the impact of this value with these or at least with one of these data set. Just to give the reader an idea on the sensibility of the parameter choice. Thus, I suggest to elaborate a bit more on this value. Just to find the impact with respect to the identified cycles. Just use the same time series, with a value equivalent to 1, 2.5, 5 and 10% of the range in y (or other values) and look for changes in the result. Maybe the results will be different with a change in the growth rate, proxy amplitude or signal to noise ratio of the smoothed record.

But then again, as the approach is semi-automatic, the user can correct anything the program found too much or too less. This should be also mentioned, when doing the sensitivity analysis.

Response: The updated manuscript provides a clearer explanation of what this hyperparameter is and how it is derived (please see extract below and Comment #3). Depending on the signal, the automatic value (2.5% of the proxy's central 95% range) can be too low or high, which was the case in sections 3.2.1 and 3.2.2, respectively. The sensitivity to this criterion depends on the signal's noise level compared to peak height. The sensitivity of a given output to this hyperparameter is now more clearly visualised in the GUI and is shown for each of the three examples (see Comment #10).

Lines (122-125)

Peaks in the inverted signal correspond to points that exceed their immediate neighbours and surpass a minimum prominence criterion. This threshold suppresses false detections caused by remaining signal noise and allows the user to manually set the value or automatically assigns it a value equivalent to 2.5% of the range in y between the 2.5th and 97.5th percentiles.

**Comment #12**: Fig. 2: I suggest to put the red line on its own y-axis (right hand side?). Otherwise it is confusing with respect to equation 1.

Response: We have changed Fig. 2 to have the template on a separate y axis.

Figure 2: A schematic overview of the matched filtering technique. (a) The Gaussian template used for matched filtering, which is set to the number of datapoints corresponding to half the estimated average cycle length of the hypothetical dataset. (b) The hypothetical dataset (grey) and the convolved matched filtering output (blue). (c) The record in (b) with the detected cycle boundaries (vertical black dashed lines) and the inserted boundaries during gap treatment (vertical green dashed lines).

**Comment #13**: Section 2.3: I agree, that this is an option to try to estimate the uncertainty. But to me it reads rather as an unusual one, as it is not the time series/signal, which is uncertain. It is the counting through the choice of the parameter set.

Therefore, I would have tried to randomly change the configuration of parameter set, the user is free to choose (or the default values). At least within a certain environment.

In this way, the signal remains as measured, as it is most likely not the signal, which is loaded with uncertainty, but rather the choice of parameter values.

I know, I would ask much, if I would request to chance the uncertainty algorithm. Probably, this would require some (heavy?) recoding. Therefore please, decide for yourself, how you like to proceed. In any way, I also provide a few thoughts on the present algorithm.

Response: We thank the reviewer for this suggestion. Our approach to estimating uncertainty is unusual and varying the hyperparameter choices is another viable option. This alternative method would only need to vary template width and minimum prominence, because average cycle length and tuning window are only used to derive the default values and adjust boundaries, respectively. Additionally, because the template width is inherently flexible (please see detected boundaries shown in Fig 8a, 10a, 12a and Comment #10) and should be based on an accurate estimate of the average cycle length, the minimum prominence is the primary source of uncertainty. Tuning this parameter by inspecting the output reduces the uncertainty and constrains the value but there is still a range of plausible values. However, this approach would need to define the environment in which to vary this parameter and that would require user-input given the wide range in signals CYCLIM could be used on. This could risk users defining their own levels of uncertainty, forgoing its objectivity. Because the current method works without specified values, we have chosen to keep the current method. However, this method has been refined (please see Section 2.3 and Comment #21) and the influence of the hyperparameters on uncertainty is now discussed in text: (Lines 177-185)

The noise level is determined using a piecewise linear fit with three breakpoints (Fig. 3b). Signal degradation with respect to noise follows an S-curve, thus having two breakpoints, but using the first would overestimate signal stability, especially in records with higher internal noise. Under these conditions the reproducibility of the reference boundaries deteriorates more linearly, suppressing and/or delaying the first S-curve breakpoint. Thus, the F1 score curve is padded, introducing a third breakpoint which finds where reconstruction fidelity first falls beyond that tolerated by the padding. This new first breakpoint derives the noise level for Monte Carlo simulation. However, because template width tends to be flexible given its use for smoothing, the threshold noise level depends on the sensitivity of the minimum prominence criterion. Therefore, if the algorithm is too sensitive to this hyperparameter choice the first breakpoint will be detected at noise levels of

Figure 3: A schematic overview of algorithmic uncertainty calculation using a dataset of 1000 noisy sine waves. (a) The algorithm's ability to reproduce the same list of minima under increasing noise levels. (b) The breakpoint locations determined from a piecewise linear fit of four segments on the F1 score curve shown in (a). (c) The variance in number of minima detected with depth under the breakpoint level of noise. The 95% confidence interval for 100 random seeds is shown relative to the automatic age model. Note that the asymmetry is the product of signal noise and highlights the direction of uncertainty.

**Comment #14**: L142: "For multiple random seeds ... in 1% increments (from 1% to 100%)." For how many multiple random seeds? Does this mean you tried e.g. 1000 random seeds and increased the white noise from 1 to 100% for each random seeds in a way that you get some statistics for your F1 approach? Please also elaborate on, what the '%' unit means in this relation. I have no idea.

Response: The number of random seeds is user-specified and tells CYCLIM how many random arrays to generate. These arrays are generated to be of equal length and standard deviation to the proxy signal to ensure noise is realistic. They are then linearly mixed into the proxy signal in increasing amounts (i.e., as a ratio from 1% to 100% noise) and the algorithm is run on each array at each increment. The new detections are compared with the clean signal to derive an F1 score. We have made this clearer in text: (Lines 158-160)

Then, for a user-defined number of random seeds, arrays of Gaussian white noise are generated of equal length and standard deviation to the proxy signal. These random signals are then progressively linearly mixed into the proxy signal from 1% to 100% (in 1% increments) to simulate random error.

**Comment #15**: L145: "that are true". What is meant by this? 'True' as in 'with respect to what was found in the undisturbed signal' (which does not mean that it is really true)?

Response: We have changed this to read: (Lines 162-163)

here the fraction of output detections that are found in the unperturbed signal

**Comment #16**: Fig. 3d and Line 175-176: I understand your argument for the stepped nature of the error estimation. I also understand, that the way the age uncertainty is calculated, floating values are produced. But to my understanding the 'deviation from mean model years' can only be integer values. Either one identified minimum counts or does not count. Is it an option to at least round the values?

Response: The floating values are the product of averaging over realisations. Once the breakpoint level of noise is determined, the proxy signal is mixed at this level with a user-specified number of random seeds. The algorithm is then run on each of these mixed signals and lists of detections are extracted. Uncertainties are then derived by percentiles of the resultant ensemble of age models, which produces floating values. We have clarified the uncertainty methodology in Section 2.3.

**Comment #17**: Fig. 7c (same in Figs 9c and 11c).: about the "temporal offset". How can the offset be floating values? I don't understand this. Please explain. To me, showing the floating values suggest that it is possible to differentiate seasons! Is that intended?

Response: The temporal mismatch/offset is here defined as the difference in age values for a given point between the published age model and the CYCLIM output. Floating values are therefore induced by slight misalignments of cycle boundaries and the age model interpolation. We have changed the figure captions of Fig. 7c, 9c, and 11c to include the following:

The temporal mismatch: the difference between the published age and either the automatic or manual model outputs at a given point. Positive (negative) values denote older (younger) ages in the published model.

**Comment #18**: Fig. 8 (same in 10 and 12): At least for a) the quality of the figure must be improved to see the differences, when someone is zooming in to see the differences in the ages. In the present manuscript version the resolution is too poor. Maybe use vector graphics?

Response: We thank the reviewer for bringing our attention to this and attach high resolution vector graphic copies of the figures to the resubmission.

**Comment #19**: Line 313: "other method" Please name this other method. Is this method mentioned in the methods-section? I can't find it.

Response: We have clarified this point, changing the text to read: (Lines 391-393)

Because the average cycle length estimate is derived from other methods (e.g., approximated from a U-Th age model or by spectral analysis), this is a fixed value leaving only the minimum prominence at the discretion of the user to change.

**Reviewer #2 comments:**

Comment #20: Forman and Baldini present a new algorithm/software for automated cycle counting. The intended application is to climate archives, such as sediments, ice cores or speleothems, but it seems that it could also be used in other research fields. The software allows the user to manually optimise the counting once the automated output is available, which is an advantage to previous algorithms, which – as far as I know – didn't offer such a feature directly in the software. I also really like that the software tries to automatically assess the uncertainty of the counting using a Monte Carlo approach.

Response: We thank the reviewer for the positive comments and their support of the methods.

Comment #21: The paper is interesting and well written, and I recommend publication in CP after minor to moderate revision. I have two general comments that should be addressed in a revised version. (i) It is not entirely clear to me how the uncertainty of the combined output (i.e., the automated and manually corrected counting) is calculated. Is the uncertainty of the automated output just ignored after 'tuning' because the manually obtained result is assumed to be both more precise and accurate than the automated output? Is the uncertainty of the automated output propagated to the uncertainty of the combined output? Does the algorithm inform the user if the manual tuning would require adjusting the automated counting far more than suggested by the

uncertainty (i.e., the automated counting has an uncertainty of 10 years, but the used wants to add 50 'missing' years)? This should be clarified and included in more detail in the discussion.

Response: Algorithmic uncertainty is approximated using a noise-based Monte Carlo approach. First, for a user-set number of random seeds, the proxy signal is combined with white noise using linear mixing from 1 to 100% in 1% increments. The algorithm is re-run at each noise level for each seed, and the new lists of minima are compared to the original using the F1 score. The first signs of disproportionate algorithmic accuracy degradation in response to increasing noise level is then used to approximate the epistemic uncertainty of manual counting. This threshold was originally determined by smoothed 2nd order derivatives to find a tipping point but has been updated to a breakpoint approach (using a piecewise linear fit; please see Comment #13 for an example). We changed this part of the methods because the first minimum often depended on the degree of smoothing and for signals with high noise levels there is not always a clear tipping point. The new method pads the F1 score curve and finds the statistically significant breakpoint. Monte Carlo simulations are then run at this noise level and the distribution in derived age models yields the uncertainties.

Previously the uncertainty was kept as the deviations from the mean and then simply transferred to the automatic and manual models. However, the updated approach shows uncertainties relative to the automatic model. This change highlights areas of the automatic model that are less certain and enables asymmetric uncertainties helping users to understand the source of the uncertainty. Additionally, because the automatic output is not always accurate (for example when the signal-to-noise ratio is low), manual tuning is not limited to be inside the uncertainties of the algorithm output. Thus, the algorithmic uncertainties are no longer mapped onto the manual chronology.

We have changed Section 2.3 to be clearer and follow the updated approach outlined above.

Comment #22: (ii) I would also like to see a more detailed discussion of the uncertainty of automated vs. manual layer/cycle counting chronologies in general. I know that this is also kind of a philosophical question, but the authors mention by themselves that their algorithm provides an 'objective' way to determine counting chronologies and the corresponding uncertainty. Even if I agree that expert knowledge of the corresponding climate archive will help to improve the automated counting chronologies, these are at the same time always affected by some degree of subjectivity. In the worst case, the user already 'knows' where a specific peak in their proxy time series should occur (e.g., at the 8.2 ka event, at the YD, etc.) and may get the impression during counting that they count too many/not enough cycles. Thus, I would really like to see an extended

discussion of the pros and cons of automated vs. manual counting, the potential effects of subjectivity and how to deal with the corresponding uncertainty. In other words, what is better? An automated counting chronology with an uncertainty of 10 years, or a manually refined chronology with a lower uncertainty (however this was determined, see my first comment) that is potentially affected by the subjectivity of the person doing the counting/tuning.

Response: We thank the reviewer for this suggestion and have included a section (4.1) in the discussion that compares manual and automated counting and under what conditions should each be chosen. The added text reads: (Lines 366-386)

The automatic output has three main advantages over counting by inspection: (1) it returns results much faster; (2) the results are reproducible with the same inputs; and (3) it reduces subjectivity. Although manual counting is more flexible and able to adapt to changes in growth rate or noise levels, it risks false positives given the position of boundaries under these conditions often becomes subjective. Similarly, if the depth-proxy signal has been compared to previous records prior to age-model development, then the inspector risks incorporating confirmation bias into the cycle count. One of the underlying assumptions of the results is that the published age models are correct; however, there is inherent uncertainty with these chronologies too. Thus, the comparison statistics gauge the accuracy of the output, but once discrepancies fall within the original age model's uncertainty, further alignment may become erroneous.

Whether to tune to automatic output thus depends on the record's structure. If the proxy shows clear, consistent cyclicity then it could be argued that the automated output is accurate, and tuning may introduce false precision. Conversely, if the waveform changes shape substantially throughout the record (e.g., due to fluctuating growth rates or noise levels), the Gaussian template may not reliably amplify annual scale cyclicity, risking systematic bias. In this case, the automated output and its accompanying uncertainties may be inaccurate and thus require manual tuning.

It is important to note that only the automatic output yields quantified algorithmic uncertainty. Manual tuning forfeits this benefit, and introduces its own uncertainty tied to the inspector's judgements. Thus, when the algorithm can only provide a basis to count from due to systematic error, the user must manually tune the output. However, when the cyclicity is sufficiently predictable to yield an accurate chronology, whether to tune the output becomes a choice. A user may decide to report both an accurate automated chronology with uncertainty and show the precise, manually tuned age model for comparison. This way small potential errors made by the algorithm can be corrected whilst also retaining the benefits of quantified uncertainty and transparency of an automated count. Regardless of the signal's clarity, any automated output should always be quality-checked to ensure cyclicity is being correctly tracked.

**Comment #23**: Title: As mentioned in my general comment, the algorithm could in principle not only be used for climate reconstruction. Thus, I would delete 'for palaeoclimate reconstruction' from the title. If the authors prefer to keep the reference to palaeoclimate, I would use 'climate archives' instead..

Response: We thank the reviewer for the suggestion. CYCLIM could certainly be used on a wide range of signals, but given the GUI and features are designed to construct age models for palaeoclimate reconstruction we have changed the title to:

CYCLIM: a semi-automated cycle counting tool for generating age models and palaeoclimate reconstructions

**Comment #24**: Line 66: 'By facilitating both efficiency and supervision, CYCLIM enhances the speed, accuracy, and transparency of cycle counting.' Even if I generally agree, I really would like to see a more detailed discussion of the potential uncertainties of automated vs. manual counting (in particular in terms of subjectivity) and how to combine them (see my general comment).

Response: Please see our response to Comments #21 and #22.

**Comment #25**: Section 2.1: This section gives a useful overview of the software, but when I read it, I immediately had many questions (e.g., about the calculation of the uncertainty). Thus, I would suggest removing all references to the technical aspects (e.g., that 'the algorithm derives a median age model from 2,000 Monte Carlo realisations using piecewise cubic Hermite interpolating polynomial (PCHIP) interpolation ...') and just describe the structure of the algorithm here.

Response: We have revised Section 2.1 in line with the reviewer's comment. Technical details have been reduced to present a clearer summary of CYCLIM's functionality. Where methods related to detection and uncertainty are mentioned, the relevant sections (2.2 and 2.3) are referenced to guide the reader to more detailed explanations.

**Comment #26**: Line 128: 'To maintain accuracy and avoid undercounting, …' I guess this paragraph describes the 'gap treatment' referred to elsewhere in the manuscript. Even if I think that I understand what the algorithm does here, it may be good to demonstrate this showing an (maybe artificial) example. This would be very helpful for the reader to understand the functionality of the algorithm.

Response: We have made the gap treatment methodology clearer using the reviewer's suggestion to include an artificial example (see Figure 2c or Comment #12).

**Comment #27**: Line 188: '... and low noise level (Fig. 3).' Here and elsewhere in the paper, please check the numbering of the figures. It seems that this should be Fig. 4.

Response: Corrected.

**Comment #28**: Fig. 5: I didn't find a reference to Fig. 5 in the text.

Response: Corrected.

**Comment #29**: Line 232: 'The automatic age model achieves a good match with that published by Kuhnert et al. (1999), albeit with clear signs of overcounting (Fig. 7a and c).' In the light of my general comment regarding the pros and cons of automated and manual counting, why should the published chronology be better than the automated counting? I am sure there are very good reasons for this assumption, but they should be mentioned and discussed in the manuscript.

Response: This study assumes the published chronologies are correct to enable statistical comparisons between the generated and original chronologies. However, these statistics should be viewed as comparison metrics to quantify general accuracy and not errors, and the text has been adapted accordingly. This assumption is now discussed in text: (Lines 370-373)

One of the underlying assumptions of the results is that the published age models are correct; however, there is inherent uncertainty with these chronologies too. Thus, the comparison statistics gauge the accuracy of the output, but once discrepancies fall within the original age model's uncertainty, further alignment may become erroneous.

Comment #30: Line 233: 'The overcount increases with depth to a maximum of 8.05 years (mean absolute error = 4.56 years, 2.3% of the target's temporal range), …' I am not sure that I completely understand the meaning of the 'mean absolute error'. Is this the mean deviation from the mean model at the bottom of the chronology shown in Fig. 3? What would be the corresponding 95%-confidence interval at the bottom of the chronology? Are these limits always symmetric (i.e., is the probability for over and undercounting always comparable)? This information is essential to assess the performance of the algorithm. If the maximum uncertainty (at the 95%-level) is 4.5 years, then an overcount of 8 years would be inaccurate.

Response: The mean absolute error was the mean absolute discrepancy between the CYCLIM age model and the published one. This has now been renamed to be 'mean absolute deviation' as the reviewer rightly mentions that the published age model should not be considered true and so it is not an error. This has been changed across the whole manuscript.

Regarding the uncertainty and tuning please see our response to Comment #21.

**Comment #31**: Line 258: 'This leads the record to have an accumulating mismatch which rises to a maximum of 8.23 years by the end of the record at 207 years counted (mean absolute error = 6.08 years, 2.8% of the targets temporal range).' Same question here. What does the error mean? Later, it is said that 12 cycles were added in the tuning stage. This is more than the uncertainty of 6 years. Does this mean that the initial automated counting was inaccurate?

Response: Please see our responses to Comments #21 and #30.

**Comment #32**: Line 266: 'At maximum depth, the 95% algorithmic uncertainty is approximately ±2 years compared to the published ±3 years.' How can the uncertainty of the tuned output be so much smaller than that of the initial automated output if 12 years were added. Are these assumed to absolutely certain even if the algorithm did not identify them? This also refers to my first general question regarding the treatment of the combined errors after tuning.

Response: Please see our response to Comments #21.

**Comment #33**: Section 3.2.3: Again, please provide a more detailed description of the different uncertainties mentioned in the text and how they were determined (in particular after tuning).

Response: Please see our response to Comments #21.

**Comment #34**: Line 308: 'The CYCLIM algorithm extracts cyclicity information from three examples both accurately and quickly via an automated matched filtering technique, ...' Does this mean that the automated chronologies were accurate within their uncertainties? This did not seem to be the case to me, but maybe I misunderstood the quoted uncertainties (see my comments above).

Response: Please see our response to Comments #21. The accuracy mentioned here was supposed to correspond to CYCLIM's ability to find most cycle boundaries automatically, but the reviewer is correct that the automatic outputs of the previous version required tuning beyond their uncertainties and so this statement is ambiguous. We have amended this sentence to read: (Lines 388-389)

The CYCLIM algorithm extracts the positions of cycle boundaries from the three examples both accurately and quickly via an automated matched filtering technique, achieving close approximations of the original chronologies.

**Comment #35**: Line 320: 'The results from this stage follow the original chronologies very closely and further running could yield a perfect match.' This again somehow suggests that the published chronologies were somehow perfect. Why should this be the case? Couldn't it be that the chronologies determined here with the (less subjective) semi-automated method are more accurate?

Response: Please see out response to Comment #29.

**Comment #36**: Line 327: '(1) reporting an objective cycle count (i.e., the algorithm output) before correction; ...' This is exactly what I consider as one of the greatest advantages of the algorithm. Thus, it is very important to better describe how the uncertainty of these 'objective' cycle counts are afterwards propagated to the final 'tuned' result. Who can guarantee that the user does not add 'false' cycles or delete 'true' cycles?

Response: We appreciate the concern regarding the manual tuning stage and have adjusted the uncertainty methodology to remove mapping onto the manual output (see Comment #21). While we cannot guarantee a user is correctly tuning the result, the tracking system allows them to transparently report the number of cycles added/removed and even show their locations for validation. We have also provided discussion of manual vs automated cycle counting (see Comment #22).

**Comment #37**: Section 5 (conclusions): This section rather reads like an abstract than conclusions. I would completely delete the 1st paragraph and focus on the results here.

Response: We have changed Section 5 in line with this comment, which now reads: (Lines 411-421)

Here we present a new semi-automated Python-based application (CYCLIM) for deriving chronologies from annual-scale cycle counting. The algorithm performs an initial cycle count using matched filtering, after which the user may inspect and refine the output by

removing false detections and/or adding in missed cycles. The method accommodates multiple forms of chronological constrains, such as an anchor point and other tie points. Based on the testing presented here, CYCLIM generates quick and accurate automatic counts, often within counting errors and an order of magnitude faster than by inspection. It provides a reliable first-pass model while also allowing users to correct the output where needed within a user-friendly GUI. When the signal is sufficiently robust to the noise-based perturbation approach, CYCLIM successfully quantifies algorithmic uncertainty, yielding confidence intervals consistent with published age errors and highlighting where cyclicity becomes more subjective. This framework thus promotes rapid, reproducible, and transparent cycle counting and can serve as a basis for reporting objective and subjective cycle boundaries, as well as developing average manual age models.